# Clostridial Infections in Cattle: A Comprehensive Review with Emphasis on Current Data Gaps in Brazil

**DOI:** 10.3390/ani14202919

**Published:** 2024-10-10

**Authors:** Felipe Masiero Salvarani, Eliel Valentin Vieira

**Affiliations:** Instituto de Medicina Veterinária, Universidade Federal do Pará, Castanhal 68740-970, PA, Brazil

**Keywords:** botulism, tetanus, blackleg, malignant edema, enterotoxemia

## Abstract

**Simple Summary:**

Clostridial infections in cattle are a major veterinary concern in Brazil, significantly impacting the country’s livestock industry. These infections are caused by various *Clostridium* species, which are anaerobic, spore-forming bacteria capable of producing powerful toxins. The main clostridial diseases affecting cattle in Brazil include botulism, tetanus, blackleg, malignant edema and enterotoxemia. Prevention and control measures of regular vaccination programs are crucial in preventing clostridial diseases. Ensuring proper hygiene, especially during surgical procedures, and minimizing injuries can help reduce the risk of infection. Surveillance and research are essential for understanding the epidemiology of these diseases and improving control strategies. Monitoring outbreaks and investigating new cases help in adapting vaccination protocols and management practices. The economic impact of clostridial infections causes significant economic losses in Brazil’s cattle industry due to high mortality rates, decreased productivity, and the costs associated with vaccination and treatment. Effective prevention and control strategies are crucial to mitigate these losses and ensure the health and productivity of cattle herds.

**Abstract:**

Clostridial infections in cattle are a significant concern for Brazilian livestock. These diseases are caused by various species of *Clostridium*, which are known for their ability to produce potent toxins. Botulism in cattle is a serious and often fatal condition caused by the ingestion of neurotoxins produced by *C. botulinum*. This bacterium thrives in decomposing organic matter, such as spoiled feed, carcasses, and contaminated water. Tetanus is less common, but it is a serious disease that follows the contamination of wounds with *Clostridium tetani* spores. It results in muscle stiffness, spasms, and often death due to respiratory failure. Blackleg (*C. chauvoei*) is a disease that primarily affects young cattle, leading to acute lameness, swelling, and high fever. Malignant edema (*C. septicum* and others) is characterized by rapid onset of swelling at wound sites, and it can occur after injuries or surgical procedures. Enterotoxemia is triggered by the rapid growth of *C. perfringens* in the gut following excessive carbohydrate intake. This leads to toxin production that causes sudden death. In conclusion, clostridial bovine infections remain a persistent challenge for Brazilian cattle farmers. With continued focus on vaccination, good management practices, and research, the impact of these diseases can be minimized, safeguarding the livestock industry’s economic viability.

## 1. Introduction

The genus *Clostridium* was first described by Prazmowski in 1880, and since then, more than 225 species distributed in different geographic areas have been identified as *Clostridium* sp. They are Gram-positive rods, sporulating and strict anaerobes, the majority of which are part of the intestinal microbiota of animals and humans, but only a few species are capable of causing diseases in animals [1,2].

Many of the infectious processes and poisonings that affect domestic animals are caused by bacteria of the genus *Clostridium*. These diseases are called clostridioses and have high mortality rates. Due to their high sporulation capacity, bacteria of this genus are capable of remaining potentially infectious in the soil for long periods, representing a significant risk to the animal and human population. Even though they are capable of producing disease in animals and humans, they are rarely considered zoonotic agents [3,4,5,6].

The pathogenic bacteria that make up this genus cause diseases basically through two mechanisms: the production of toxins and tissue invasion. Clostridia enter the body in sporulated form through contaminated food, wounds or by inhalation. The toxins are produced in the animal’s body or are ingested pre-formed. Among the toxins of clostridial origin, botulinum and tetanus neurotoxins and epsilon toxin produced by *C. perfringens* types B and D stand out as the most potent toxins of known microbial origin [7,8,9].

Clostridiosis is one of the main diseases that affect domestic animals in the country, with high morbidity and lethality rates, causing great economic losses to the production sector [10,11]. The main agents involved, and the diseases caused by bacteria of the genus *Clostridium* in cattle are presented in Table 1. This review covers the main clostridioses that affect cattle in Brazil and their impacts on one of the largest cattle producers and exporters in the world.

## 2. Botulism

Botulism, derived from the Latin botulus, which can be translated as sausage, was first described in 1820 in Germany after several cases of flaccid paralysis in humans associated with the ingestion of sausages and meat sauces. Initially considered to be a fungus, it was only between 1895 and 1897 that it was demonstrated that botulism was caused by the toxin of an anaerobic bacillus known as *Bacillus botulinus*. During this same period, using a cell culture filtrate free of bacilli and spores, signs of paralysis were reproduced in laboratory animals, confirming the existence of a toxin [12,13,14].

Currently called *Clostridium botulinum*, this anaerobic bacillus is classified into seven types (A, B, C, D, E, F and G) based on the antigenic specificity of the neurotoxin (BoNT) produced by each strain. Encoded by the BoNT gene, these toxins are produced as 150 kDa polypeptide chains, which are cleaved into two smaller chains prior to release by the microorganism. Therefore, a heavy chain (HC) of 100 kDa and a light chain (LC) of 50 kDa are created, which remain linked through a disulfide bond. The origin of the BoNT gene varies between types of *C. botulinum*: types A, B, E and F, which cause human botulism, are of chromosomal origin; types C and D, responsible for the vast majority of cases of the disease in animals, originate from bacteriophages; for type G, the origin is a plasmid. Interventional studies involving animals or humans and other studies that require ethical approval must list the authority that provided approval and the corresponding ethical approval code [1,5,8,12,13,14].

The mechanism of action of BoNTs (Figure 1) is endowed with three steps: binding, translocation and enzymatic activity. In the first stage, HCs bind to the membranes of neurons, mainly cholinergic, through a double-receptor system consisting of a ganglioside and a protein component. Then, BoNTs are translocated to the neuronal cytoplasm via endocytosis. It is believed that, during this step, HCs form pores in the membrane, through which LCs pass from the extracellular to the neuronal intracellular environment. Finally, LCs cleave one or more SNARE proteins, which are responsible for the docking and fusion of vesicles containing neurotransmitters at presynaptic terminals. As a result, there is a reduction in the release of acetylcholine at neuromuscular junctions, which leads to the inability to contract muscles or flaccid paralysis of skeletal muscles. Each serotype cleaves specific peptide bonds from one or more SNARE proteins [1,5,6,8,12,13,14].

Animal species have different susceptibilities to BoNTs and botulism. Horses appear to be sensitive to all serotypes. Cattle are susceptible to types C and D, rarely affected by types A and B, presenting acute or subacute conditions. The disease in sheep is commonly chronic. Dogs are less sensitive, with botulism being caused by serotypes A, B and C, but rarely. Birds are affected by types C and, more rarely, A and E, although they can eliminate all serotypes in their waste [1,5,6,8,12,13].

Botulism in cattle was first described in Brazil in 1970, in Piauí, and 13 years later in Rio Grande do Sul. From then on, botulism in domestic ruminants began to occur in an epizootic form. Animals with high nutritional requirements, such as pregnant or lactating females, raised in soils and pastures poor in minerals, especially phosphorus, without adequate mineral supplementation, developed the habit of osteophagy (eating bones) or sarcophagy (eating corpses), seeking to supply its mineral deficiencies. However, at the same time, they ingested pre-formed BoNTs during the decomposition of carcasses, which led to large epidemics of the disease with the death of thousands of animals [13,15,16,17,18,19].

Over the past two decades, ruminant botulism in Brazil has occurred mainly in the form of sporadic outbreaks. In these outbreaks, BoNTs ingested by animals originate from food contaminated with decomposing organic matter, such as chicken litter, hay, grains, feed and silage of poor quality or poorly stored. In addition to these, water troughs with carcasses of small animals or other types of decomposing organic matter, as well as wells and ponds with stagnant water, can serve as a source of BoNTs for animals [20,21].

The incubation period and severity of botulism will depend on the amount of toxin ingested and the susceptibility of the animal species. In ruminants, the course of the disease can last from hours to a few weeks, and lethality is close to 100%. The initial clinical signs are difficulty in locomotion and incoordination of the hind limbs, with cranial progression of flaccid paralysis (Figure 2). The animal enters a pre-agonic state, and death, preceded by a coma, occurs due to cardiorespiratory arrest. Throughout the symptomatology, the psyche of the animals remains unchanged. Injuries at necropsy are rare and limited to petechiae in the myocardium as a consequence of respiratory agony, which precedes death [12,13,16].

The diagnosis of botulism is based on epidemiological data, clinical signs of affected animals and the detection of BoNTs in clinical specimens and/or sources of poisoning. These may include rumen, gastric and intestinal contents, liver, serum, and samples of food and water that the animals may have ingested. The standard test for BoNT research is the mouse bioassay. This test is based on the intraperitoneal injection of diluted samples into mice. If the toxin is present, the mice develop typical signs of botulism, such as raised hair, muscle weakness and dyspnea, which is manifested by a narrowing of the waist, called “wasp waist”. The type of BoNT is determined by neutralization of the toxin with its specific antitoxin. Although the mouse bioassay is highly specific and sensitive, with detection limits of up to 0.01 ng/mL of sample, other “in vitro” tests are available or under development. Highlights include enzyme-linked immunosorbent assays (ELISA), polymerase chain reaction (PCR), real-time PCR, chemiluminescence, electrochemiluminescence, radioimmunoassay, lateral flow immunoassay, endopeptidase assay and complement microfixation [1,5,15,19,21].

Vaccination with *C. botulinum* toxoids C and D is the main way to control botulism in domestic animals, with the only exception being birds. In association, other measures are extremely important, especially those that aim to prevent the ingestion of pre-formed BoNTs by animals. These measures include adequate mineral supplementation of domestic ruminants; removal of mammal and bird carcasses from pastures and watercourse edges; adequate production, storage and supply of high-quality food; not providing poultry litter to domestic ruminants under any circumstances; providing good quality water and preventing animals from accessing places with stagnant water or water of unknown quality [10,11,22,23].

Although current data on the epidemiology of Clostridiosis are scarce, botulism in cattle is the main observed disease caused by the genus *Clostridium*. The water form of botulism, caused by the ingestion of contaminated water and via osteophagy due to the lack of mineral supplementation in cattle, is the primary way in which the disease occurs. Despite the availability of efficient commercial vaccines against botulism, there is a lack of awareness among rural producers about the use of these vaccines. Additionally, Brazilian veterinarians often do not take the initiative to send biological material suspected of botulism to laboratories for investigation. Unfortunately, there are few laboratories in the country equipped to handle such cases.

## 3. Tetanus

Tetanus is a non-contagious infectious disease caused by the action of exotoxins produced by *C. tetani*, which cause functional changes in the central nervous system with increased excitability. *C. tetani* is cosmopolitan and is commonly found in soil in the form of spores and the intestinal tract of humans and domestic animals. Infection generally occurs through contamination by spores on the skin or mucosa with superficial or deep lesions of any nature, but mainly surgical castration, castration with bands associated with tetanus. Improperly cutting and healing navels is a potential entry point for *C. tetani* spores in large animals. Factors such as the presence of devitalized tissues, foreign bodies, ischemia and infection contribute to the reduction in the redox potential in the lesion, which favors the germination of spores that multiply and produce the toxins tetanolysin and tetanospasmin, the latter being responsible for the clinical characteristics of tetanus [24,25].

Tetanospasmin is a potent neurotoxin encoded by the TeTx gene of non-conjugative plasmid origin. It is produced during the stationary phase of growth as a polypeptide with a molecular weight of 150 KDa composed of a heavy chain (HC) and a light chain (LC) linked via a disulfide bond. HC has a binding domain in its C-terminal region capable of recognizing specific receptors in the presynaptic terminals of the central nervous system. The toxin is internalized and transported in endosomes to the cell body of the motor neurons located in the spinal cord. Then, acidification of the endosome occurs, which results in a conformational change in the N-terminal domain of the HC, followed by its insertion into the endosome membrane and passage from the LC to the cell cytosol. Once in the cytosol, the LC is capable of cleaving SNARE proteins (soluble N-ethylmaleimide-sensitive factor attachment protein receptor) responsible for exocytosis in neurons. As a result, there is a reduction in the release of inhibitory neurotransmitters such as gamma globulinic acid and glycine, resulting in spastic paralysis (Figure 3). Tetanolysin is a hemolysin capable of causing cell lysis through the formation of pores by hydrolysis of plasma membrane phospholipids. The clinical significance of this enzyme is unknown; however, it is inhibited by plasma oxygen and cholesterol [6,24,25].

In Brazil, tetanus outbreaks causing cattle mortality have been identified in all Brazilian states [6,26,27,28,29,30,31,32], and clinically, the disease manifests itself with low or absent fever and muscular hypertonia, causing jaw trismus, neck stiffness, protrusion of the third eyelid, dysphagia, hyperextension of limbs, opisthotonus (Figure 4) and respiratory failure. Facial muscle spasticity (sardonic laughter) commonly observed in humans has already been described in dogs and cats. Initially, spasms and paroxysmal contractures are caused by tactile, sound, light stimuli or high ambient temperature, and as the disease progresses, they can be triggered spontaneously. In general, the animal remains conscious.

The diagnosis of tetanus is based on history, anamnesis and clinical signs and does not depend on laboratory confirmation. At necropsy, no significant lesions are found, and the presence of an entry point for the agent is not always observed, as in some reports of outbreaks of idiopathic tetanus in young cattle. Differential diagnoses for poisoning by metoclopramide and neuroleptics are important: strychnine poisoning with the absence of trismus and generalized hypertonia during intervals of spasms and meningitis with the presence of high fever from the beginning, absence of trismus and vomiting. In addition to these, rabies is included as a differential diagnosis, in which it is possible to observe the presence of convulsions, changes in behavior, and absence of trismus in addition to a history of biting, scratching or licking by animals [26,27,28,29,30,31,32].

The basic principles of tetanus treatment are sedation, neutralization of circulating tetanus toxin by anti-tetanus serum, debridement of the infectious focus, eradication of the agent with administration of penicillin and general supportive measures. The animal must be kept in a dark and silent environment, with a stable and pleasant temperature, in order to minimize the signs presented. Toxoid vaccination is the main form of tetanus prevention, and the commercial product is available for cattle. The administration of a booster vaccine prior to surgical procedures, as well as the administration of anti-tetanus serum at the time or after them, is essential for the prophylaxis of the disease [26,27,28,29,30,31,32]. Tetanus is the second most common clostridiosis in Brazil, despite the lack of current data on the epidemiology of this disease in cattle. The incidence of tetanus in livestock production in Brazil is closely linked to surgical practices carried out without proper asepsis. Consequently, the use of anti-tetanus serum is a common practice adopted by producers and veterinarians. However, this practice often does not prevent the occurrence of tetanus, even though a commercial vaccine is available. The vaccine is not typically used by rural producers and veterinarians, who often cite cost issues as the reason for their non-use. However, starting in 2024, Brazil will be free of foot and mouth disease without vaccination, which could boost the use of clostridial vaccines. Only time will tell if this expectation will be realized.

## 4. Histotoxic Diseases

Blackleg is an acute disease of cattle and sheep caused exclusively by *C. chauvoei*. In cattle, the disease occurs endogenously and is frequently observed in young animals. The ingestion of spores present in the environment is the main route of transmission (Figure 5). Although most spores are excreted in feces, some may leave the intestine and be distributed to tissues where they remain dormant. The sequence of events that leads to the distribution of endospores in tissues is unclear, but it is believed that they are transported by macrophages [33,34,35].

Gas gangrene is an exogenous and necrotizing infection of soft tissues that affects several species of domestic animals (Figure 6). This pathology is caused either by one or the association of the following species of the genus *Clostridium*: *C. septicum, C. chauvoei, C. novyi* type A, *C. perfringens* type A and *C. sordellii*. The occurrence of this disease is related to strict contact between these agents and domestic animals, which favors the contamination of wounds resulting from surgical and/or sanitary practices carried out without aseptic care. Furthermore, tissue injury promotes a decrease in redox potential, alkalinization of pH and decomposition of protein products. These factors contribute to the penetration, germination and intense proliferation of clostridia, with consequent production of toxins responsible for the pathological condition of the disease [1,8,36,37,38,39,40].

*Clostridium septicum* was the first species of *Clostridium* identified, being named *Vibrion septicum* by Pasteur and Jubert in 1877. The alpha toxin (Table 2), the main toxic factor of this agent, binds to receptors present in the cell membrane of the clostridia hosts where they are activated by proteolytic cleavage. Activated monomers diffuse laterally across the membrane, forming an oligomer that undergoes conformational changes until an active pore spanning the cytoplasmic membrane is installed. The pore promotes changes in cell permeability, followed by water influx and consequent rupture, determining cell lysis and cytotoxicity [1,8,36,37,38,39,40].

Although descriptions of gas gangrene and blackleg have been made since the mid-19th century, *C. chauvoei* was only described in 1880. Unlike other bacteria that cause myonecrosis, few studies have been carried out seeking to elucidate its toxic activity and mechanisms of action of the toxins produced by *C. chauvoei*. The CctA toxin (Table 2) was recently identified as the main virulence factor of this microorganism, having the capacity to provide protection to animals vaccinated with its toxoid. However, it is considered necessary to carry out other studies to better characterize this and other proteins produced by *C. chauvoei* [1,8,36,37,38,39,40].

*Clostridium novyi* is classified into four types, from A to D, according to the production of toxins. Type A produces only alpha toxin and is the agent of gas gangrene in humans and domestic animals. Type B produces, in addition to the alpha toxin, the beta toxin. Type C does not produce toxins, and type D, now called *C. haemolyticum*, causes bacillary hemoglobinuria in cattle. *C. novyi* alpha toxin (Table 2) is a glycosyltransferase that catalyzes the glycosylation of small GTPases, determining the inactivation of cytoskeletal proteins and disorganization of actin filaments, resulting in effects such as cell rounding, loss of intercellular junctions and increased in endothelial permeability, which is compatible with the edema observed in conditions caused by this bacterium [1,8,36,37,38,39,40].

*Clostridium sordellii* was first isolated in 1922 by Alfredo Sordelli, an Argentine microbiologist who named the bacteria *Bacillus oedematis sporogenes,* and since 1928, the name *C. sordellii* has been adopted. Animals affected by *C. sordellii* tend to present less disseminated and painful lesions with vascular damage and reduced hemolysis. The virulence of this bacterium is attributed to numerous exotoxins, although only the lethal and hemorrhagic toxins are considered essential for the disease to occur (Table 2). Both catalyze the glycosylation of small GTPases, differing only in terms of target GTPases. These toxins have a glycosyltransferase action, modifying the GTPases that control the cell cycle, apoptosis, gene transcription and the structural functions of actin, such as cell morphology, migration and polarity. The induced modifications cause actin condensation, culminating in cytoskeletal disorganization, cell rounding and eventual apoptosis [1,8,36,37,38,39,40].

The first description of *C. perfringens* type A was made from necropsy isolates from a cadaver with disseminated emphysema. *C. perfringens* type A is often associated with disease in both dairy and beef replacement cattle. This *Clostridium* differs from other bacteria of the genus due to its relative aerotolerance, high and rapid growth rate and genetic instability regarding the expression of toxin-coding genes. Alpha toxin (Table 2), the main toxic factor involved in cases of gas gangrene, interacts with membrane phospholipids where it has potent phospholipase C activity, hydrolyzing the membrane of eukaryotic cells. Furthermore, this toxin determines vascular collapse due to an exacerbation in the concentrations of arachidonic acid and its metabolites in the affected cells, which leads to a local inflammatory reaction and vasoconstriction. The alpha toxin activates protein kinase C, determining the production of platelet aggregation factor with consequent formation of intravascular thrombi, a fact that contributes to inflammation and local anoxia. Luminal obstruction of capillaries by neutrophils is pathognomonic of *C. perfringens* infection. The reasons for leukostasis are unknown, but it is believed that it may be related to the necrosis of endothelial cells, which makes it impossible to rearrange the cytoskeleton that allows the transmigration of defence cells [1,8,36,37,38,39,40,41].

In Brazil, when gas gangrene or blackleg is not quickly controlled, the animal develops systemic toxemia and shock. Death occurs as a consequence of the effect of toxins on the hemodynamics of the vascular system. Clinically, the animal presents elevated or normal temperature, anorexia and depression. When a limb is hit, there is difficulty in locomotion and a variable increase in volume associated with subcutaneous crepitus due to the large amount of gas produced by bacterial multiplication in the focus of infection, edema and hemorrhage. Sometimes, these changes are intense enough that the skin appears tense, diffusely red or black, with bruises and suffusions. It is not uncommon to see a line delimiting the infected part from the healthy part. When cut, extravasation of liquid can be observed, with gas bubbles and subcutaneous hemorrhage. The fascia and muscle bundles can be separated when there is an intense amount of gas. The muscles appear intensely hemorrhagic, emphysematous and with gray areas, indicative of necrosis. Histologically, diffusely swollen, eosinophilic muscle fibers are observed, with loss of striations (hyaline degeneration) and/or diffusely fragmented (floccular necrosis), in the presence of bacilli. Inflammatory infiltration varies from absent to moderate, consisting mainly of neutrophils, with hemorrhage, edema and gas also observed in varying degrees in the tissue. Anthrax can also occur in its visceral and asymptomatic form when internal organs, such as the tongue, diaphragm or heart, are affected. Consequently, the animal has no visible injuries, and cases of sudden death are frequent [33,34,35,36,41].

Clostridia are saprophytic microorganisms that promote the decomposition of the carcass of dead animals, which is why the material must be collected immediately after the death or euthanasia of the animal. The material can be shipped fresh, cooled, frozen or fixed in 10% neutral formalin (used for preserving tissue morphology and is not suitable for maintaining bacterial load in samples intended for microbial diagnostics). The laboratory diagnosis of myonecrosis is presented in Table 3 [41].

Affected animals must be quickly treated due to the super-acute course of the disease, with intravenous administration of high doses of penicillin. The chances of recovery are greater for animals at the beginning of the infection or those whose muscle damage is not widespread, but, in general, treatments are not successful. The most important measure for controlling and preventing myonecrosis outbreaks is the systematic vaccination of the herd. The use of vaccines results in a marked reduction in the incidence of these diseases. However, it is essential that hygiene measures are adopted, such as disinfection of needles and surgical instruments, asepsis of vaccine application sites or surgical procedures, and adequate handling of carcasses, among others [8,10,11,33,42].

Histotoxic diseases are considered the third most common clostridiosis in Brazil, with gas gangrene being more frequent than blackleg, despite the lack of current data on the epidemiology of this disease in cattle in Brazil. Histotoxic diseases have an acute to super-acute infection period, meaning that animals are often found dead. For the diagnosis of histotoxic clostridia, the time between death and the collection of material for diagnosis must be a maximum of 6 h. The lack of habit among veterinarians in collecting biological material for the diagnosis of histotoxic diseases exacerbates the problem of the lack of diagnosis and epidemiological data on clostridial diseases in Brazil. The clostridial vaccines available on the Brazilian market are multivaccines, containing several clostridial antigens, including those for gas gangrene and blackleg. However, the lack of routine use of clostridial vaccines impedes the prevention of these diseases. This is an area where rural extension services must work together with public policies aimed at combating infectious diseases to reduce the losses resulting from the death of animals due to infectious diseases, especially in a country that relies on agribusiness as the basis of its economy.

## 5. Enterotoxemias

Enterotoxemias caused by C. perfringens occur under specific conditions in the presence of certain predisposing factors, such as abrupt changes in diet, overeating, intestinal stasis, or any condition that disrupts the normal gut flora. As a consequence, the agent multiplies in the gastrointestinal tract of animals and produces exotoxins, which are mainly responsible for the development of the nosological condition. *C. perfringens* is classified into five types (A–E) based on the production of four main toxins: alpha, beta, epsilon, and iota (Table 4) [5,43,44,45,46,47,48,49].

*Clostridium perfringens* type B is the causative agent of dysentery in newborn lambs and can also affect goats, calves and foals. This condition affects lambs under two weeks of age and is mainly caused by the action of beta toxin, which is inactivated by proteolytic enzymes, such as trypsin. The fact that the disease occurs primarily in newborn animals comes from excess colostrum, as colostrum is a source of antitrypsinic factors, thus favoring the action of the toxin. The disease presents a morbidity rate of around 30%, with lethality reaching 100% of affected animals. The course is peracute; generally, within a few hours, the animals present severe abdominal pain, reduced suction of colostrum and/or milk, semi-fluid and red-stained feces, in addition to developing enteritis, with extensive hemorrhage and ulceration of the small intestine [9,43].

*Clostridium perfringens* type D is the agent of enterotoxemia, commonly called overfeeding disease or pulpous kidney disease. In Brazil, there are reports of diseases in cattle causing mortality and considerable damage to Brazilian agribusiness [6,27,28,29,30,31,49,50]. Changes in the rumen microbiota as a result of sudden changes in diet, provision of diets rich in carbohydrates and low in fiber, among other factors, lead to the multiplication of the agent in logarithmic proportions, producing large amounts of epsilon toxin in the form of protoxin, being converted into a lethal protein by the action of digestive trypsin or by secondary toxins from *C. perfringens*. The activated toxin acts on the intestinal epithelium, causing increased vascular permeability, and, upon reaching the blood circulation, reaches organs such as the brain, kidneys, lungs, liver and heart, where it binds to specific receptors on endothelial cells, leading to degeneration. of these cells. With increased vascular permeability, extravasation of fluids and proteins occurs into the perivascular space, with consequent edema. When it occurs in brain tissue, it is called eosinophilic proteinaceous perivascular edema or microangiopathy [44,45,46,47,48,49,50].

In cattle in Brazil, under natural conditions and in most cases, the death of animals occurs during the first six to 18 h, but if they survive for more than 36 to 48 h, necrosis of the brain tissue occurs, known as focal symmetric encephalomalacia (Figure 7). The most common clinical form is super-acute, with death within four to eight hours. Neurological changes such as opisthotonus and pedaling movements can be observed, as well as respiratory changes such as tachypnea and pulmonary edema. At necropsy, findings may not be evident, but in some cases, hydrothorax, hydropericardium, hydroperitoneum, pulmonary edema and cerebellar hernia are found. Histopathological lesions are neurological lesions restricted to histopathological findings of eosinophilic proteinaceous perivascular edema or microangiopathy and focal symmetric encephalomalacia [6,27,28,29,30,31,49,50,51].

The confirmatory diagnosis is directly dependent on the detection of the toxin(s) produced by the agents directly in the intestinal contents. The conventional method for detecting these toxins is the serum neutralization test in mice. There are also different PCR techniques that, despite not directly detecting the toxin(s) produced by *C. perfringens* types A-E, but rather the genes that encode their production, allow the typification of the *C. perfringens* involved in a particular case or outbreak. Although the presence of toxins in intestinal contents is the most important indicator of enterotoxemia, it is necessary to combine this finding with others, especially the history of the animal and the property, clinical signs, necropsy findings and histopathology of the injured organs [49,50].

In Brazil, the control measures aim to ensure correct hygiene management, environmental disinfection and systematic vaccinations of the entire herd, as animals are in permanent contact with agents and factors that could trigger diseases. In diseases that affect young animals, female vaccination, with the aim of transferring passive immunity to their progeny, is the main strategy. In animals over four months of age, primary vaccination is recommended, with a booster dose four to six weeks later and annual revaccination [52,53,54].

Enterotoxemia in cattle due to *Clostridium perfringens* types B and D is a less prevalent disease in Brazil. In fact, there are many questions about whether the disease genuinely does not occur or if there is underreporting, as we have already discussed the lack of diagnosis. In Brazil, there was a large marketing campaign led by multinational companies on cattle enterotoxemia caused by Clostridium perfringens types B and D. However, the first step, which was to investigate whether or not these diseases occurred in Brazil, was not carried out. The clostridial vaccine, which includes multiple agents, already contained alpha, beta, and epsilon toxoids, and therefore, it was convenient to state that Brazil had enterotoxemia. Yet, in more than 20 years of work, there have been no confirmatory cases of cattle enterotoxemia caused by *Clostridium perfringens* types B and D. Nevertheless, it is reiterated that either the disease is genuinely not that common, or its epidemiological characteristics, combined with the non-collection of material for diagnosis and the scarce laboratory network for diagnosing anaerobes in Brazil, contribute to its low frequency. Therefore, new studies and research need to be conducted with a specific focus on these diseases, as enterotoxemia in cattle due to *Clostridium perfringens* is extremely frequent and important in other parts of the world.

## 6. Vaccines

Multivalent conventional clostridial vaccines are widely used to prevent clostridial diseases in cattle in Brazil. These vaccines combine multiple antigens from different *Clostridium* species. The inclusion of multiple antigens in a single formulation aims to provide comprehensive protection against various clostridial diseases, such as enterotoxemia, blackleg, gas gangrene, botulism and tetanus. The effectiveness of these vaccines depends on several factors, including vaccine quality, animal health at the time of vaccination, and adherence to an appropriate vaccination schedule. Field studies have shown that multivalent vaccines are effective in reducing the incidence of these diseases, significantly contributing to herd health and productivity [2,10,33].

In Brazil, vaccination against clostridial diseases is a common and essential practice in animal health management, especially in regions where livestock farming is a predominant economic activity. The high prevalence of clostridial diseases in various parts of the country, particularly in areas with large concentrations of cattle, makes vaccination a critical preventive measure. Vaccination programs are regularly implemented on rural properties, often as part of an integrated management protocol that includes good husbandry practices, adequate nutrition, and parasite control. Producer awareness of the importance of vaccination, along with the availability of effective vaccines, has been crucial in reducing mortality and economic losses associated with clostridial diseases [2,10,33,53,54].

In recent years, the development of recombinant vaccines against clostridial diseases has emerged as a promising area of research in Brazil. Recombinant vaccines are produced using genetic engineering techniques to express specific *Clostridium* antigens, potentially offering a more targeted and effective immune response. This type of vaccine can provide advantages over conventional vaccines, including greater safety and stability, as well as a more robust immune response. Research conducted at Brazilian institutions has focused on developing recombinant vaccines that target the main toxins produced by different *Clostridium* species, such as the alpha, beta, and epsilon toxins of *Clostridium perfringens*. Preliminary studies indicate that these vaccines can induce a strong immune response and provide effective protection against clostridial diseases, representing a significant advance in disease prevention [11,22,23,53,54].

Vaccination is a fundamental tool in preventing clostridial diseases in cattle in Brazil. Multivalent conventional vaccines have been widely used and have demonstrated efficacy in reducing the incidence of various clostridial diseases. However, the development of new recombinant vaccines offers a promising perspective, potentially improving protection against these diseases and contributing to the sustainability of livestock farming in the country. Continuous investment in vaccine research and development is essential to address the challenges of clostridial diseases and ensure the health of Brazilian cattle herds [2,10,33,53,54].

## 7. Conclusions

Clostridiosis in cattle presents a significant challenge for the Brazilian livestock industry. These infections, caused by various species of the *Clostridium* genus, are notorious for their rapid onset and high mortality rates, leading to substantial economic losses. Key clostridial diseases affecting cattle in Brazil include botulism (*Clostridium botulinum),* tetanus (*Clostridium tetani*), blackleg (*Clostridium chauvoei*), malignant edema (*Clostridium septicum* and others) and enterotoxemia (*Clostridium perfringens*). The current challenges are high mortality and economic impact because Clostridial diseases often result in sudden death, causing immediate and significant losses to cattle herds. The high mortality rate directly translates to financial setbacks for farmers due to the loss of livestock and decreased productivity. The environmental persistence of *Clostridium* spores can survive in the environment for extended periods, making it difficult to predict and control outbreaks. This persistence in soil and organic matter means that even well-managed farms can experience sudden disease outbreaks. The diagnosis and treatment are hampered because the rapid progression of these diseases often leaves little time for effective intervention. While antibiotics and antitoxins can be used in some cases, their efficacy is limited if not administered early in the disease course. Additionally, the cost and logistics of treatment can be prohibitive for many farmers. Therefore, prevention and control with regular use of vaccination remains the cornerstone of clostridial disease prevention. Vaccination programs should be comprehensive, covering all major clostridial pathogens, and should be consistently administered to young and at-risk cattle. Also, farm management practices that focus on good hygiene and management practices are crucial. This includes proper disposal of carcasses, ensuring clean and uncontaminated feed and water sources, and minimizing injuries that could become infection sites. Continuous surveillance for clostridial disease outbreaks helps in early detection and response. This requires robust veterinary services and cooperation among farmers, veterinarians, and government agencies.

Future perspectives are expected to prioritize the development of more effective and long-lasting vaccines. Specifically, research into multivalent vaccines, which could protect against multiple clostridial species with a single injection, could greatly improve vaccine compliance among farmers and lead to better disease control outcomes. In addition to vaccines, advancements in diagnostic technologies are essential. The development of rapid, on-farm diagnostic tests would enable earlier detection of clostridial diseases, allowing for timely intervention and improving animal survival rates. Portable and cost-effective diagnostic tools would be particularly beneficial for small-scale and remote farmers, who often face logistical challenges in accessing veterinary services. Education and training are also critical. Farmers need to be equipped with the knowledge of best practices for disease prevention and management. Extension services and veterinary outreach programs can play a pivotal role in spreading this knowledge. Moreover, ongoing research into the epidemiology, pathogenesis, and control of clostridial diseases is essential to gaining new insights and developing innovative solutions. Strong collaboration between research institutions, government agencies, and the private sector will be key to driving advancements in clostridial disease management in Brazil. This revision clarifies that your goal is to improve disease prevention through enhanced vaccines, diagnostics, farmer education, and collaboration across sectors. In conclusion, while clostridial diseases pose a formidable challenge to cattle farming in Brazil, a multifaceted approach involving vaccination, good management practices, surveillance, and research holds the key to mitigating their impact. By investing in future perspectives, the Brazilian livestock industry can enhance its resilience against clostridial infections, ensuring the health and productivity of cattle herds and the economic stability of the sector.

## Figures and Tables

**Figure 1 animals-14-02919-f001:**
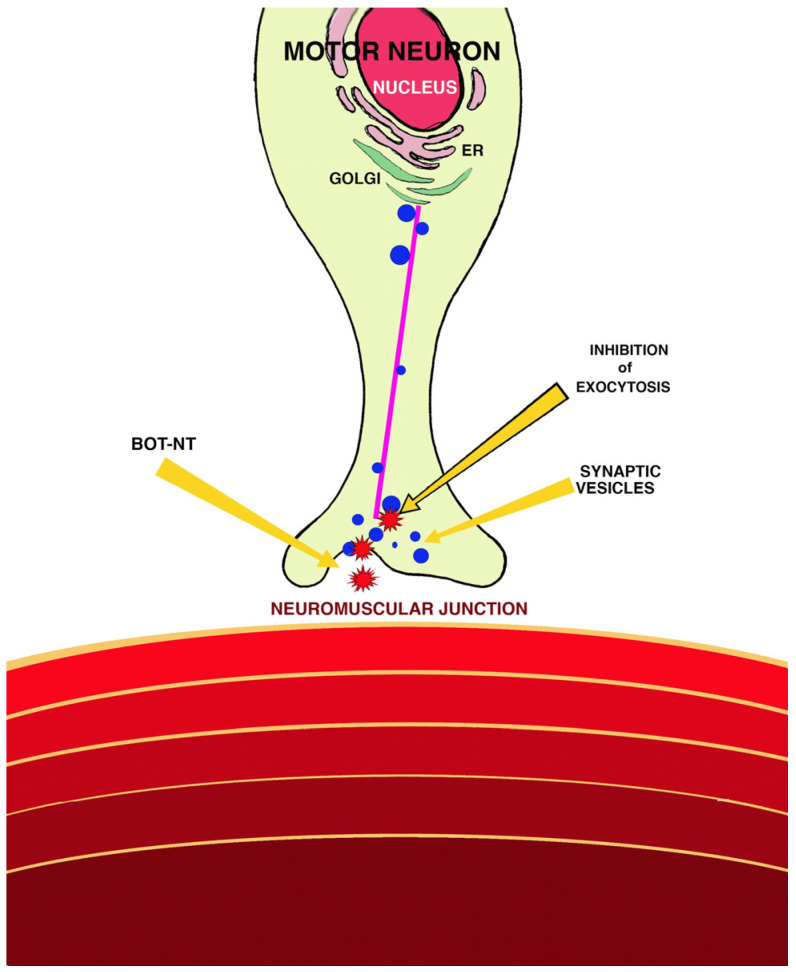
Pathogenesis of botulism [6].

**Figure 2 animals-14-02919-f002:**
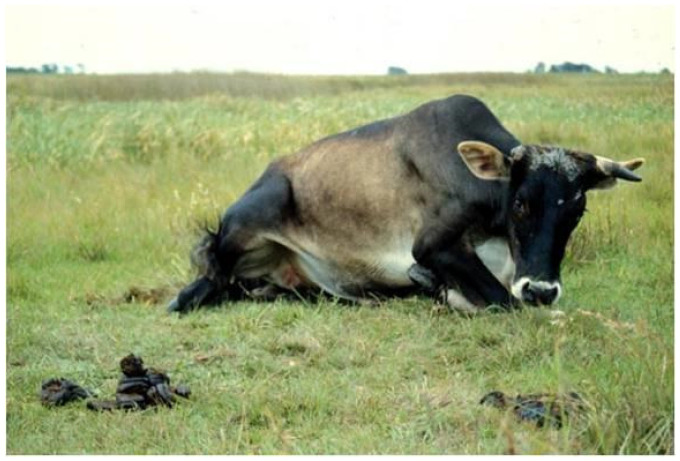
Clostridial diseases. Botulism. Flaccid paralysis in a cow. The lack of tonus in the muscle limbs (flaccidity) prevents the steer from standing [6].

**Figure 3 animals-14-02919-f003:**
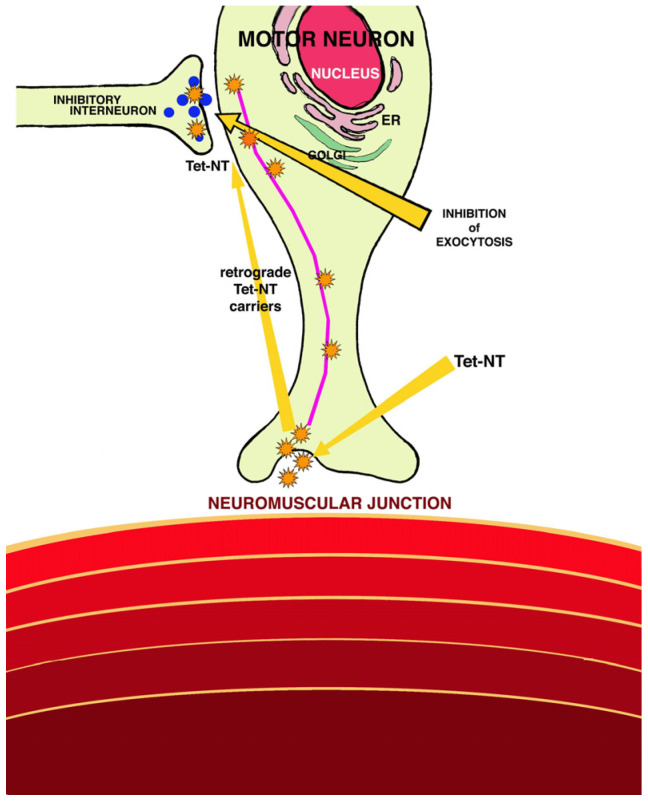
Pathogenesis of tetanus [6].

**Figure 4 animals-14-02919-f004:**
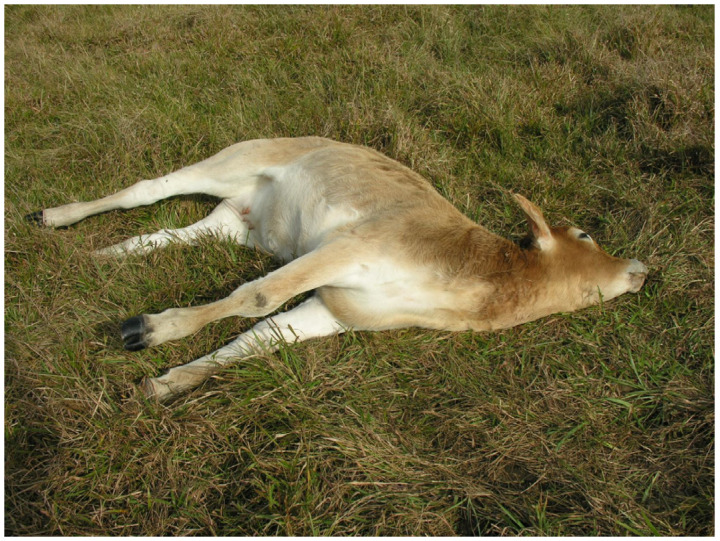
Clostridial diseases. Tetanus. Steer with opisthotonus [6].

**Figure 5 animals-14-02919-f005:**
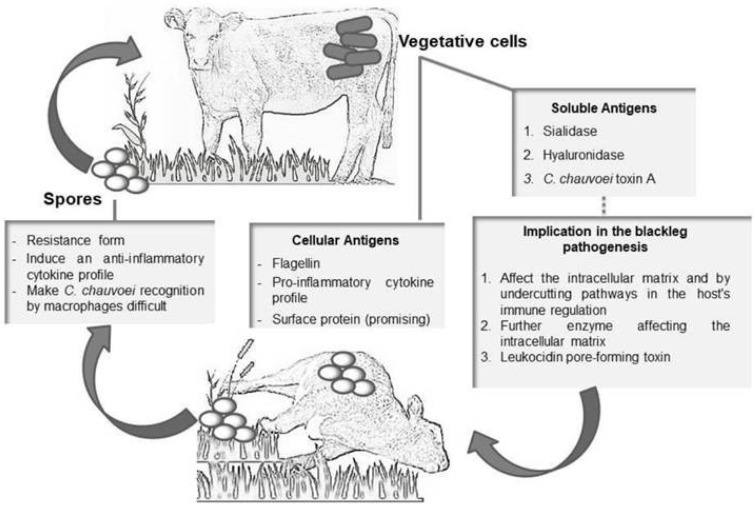
Schematic illustration of blackleg pathogenesis with currently considered major virulence factors [33].

**Figure 6 animals-14-02919-f006:**
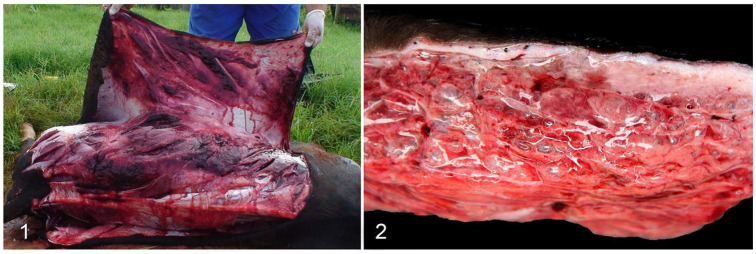
Gas gangrene is caused by *Clostridium septicum* in a heifer. (**1**) Severe subcutaneous hemorrhage and edema. (**2**) Severe subcutaneous edema expanding the subcutis [36]. Reprinted with permission from Ref. [36].

**Figure 7 animals-14-02919-f007:**
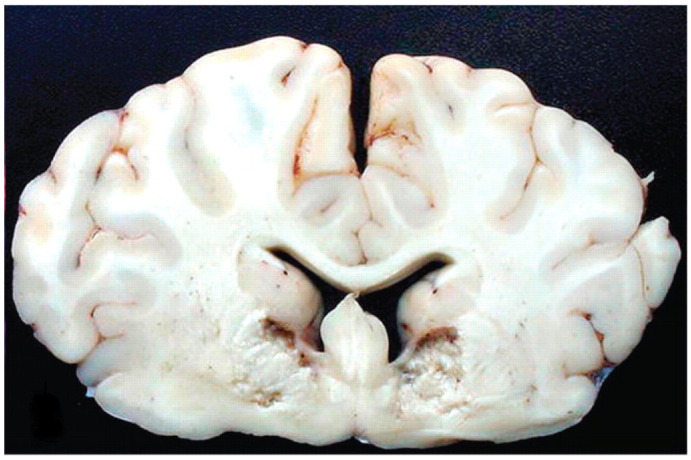
Focal symmetrical encephalomalacia. This animal developed progressive neurologic signs that became severe 8 days after inoculation, at which time the calf was euthanatized [49]. Reprinted with permission from Ref. [49].

**Table 1 animals-14-02919-t001:** Main clostridial diseases in cattle and their agents.

	Disease	Agent
Neurotropic Diseases	Botulism Tetanus	*Clostridium botulinum* *Clostridium tetani*
Histotoxic Diseases	Blackleg and Malignant edema	*Clostridium chauvoei* *Clostridium septicum* *Clostridium sordellii* *Clostridium novyi* tipo A *Clostridium perfringens* tipo A
Enterotoxemias	Enterotoxemia of Cattle	*Clostridium perfringens* tipos B and D

**Table 2 animals-14-02919-t002:** Histotoxic clostridia and characterization of the main virulence factors.

	*Clostridium septicum*	*Clostridium chauvoei*	*Clostridium novyi*	*Clostridium sordellii*		*Clostridium perfringens*
Toxin	Alpha	CctA	Alpha	Lethal	Hemorrhagic	Alpha
Origin	Plasmidial/ chromosomal	Chromosomal	Phage	Chromosomal	Chromosomal	Chromosomal
Molecular weight (kDa)	48	33	250	300	260	42
Action	Pore formation	Pore formation	Inactivation of GTPases	Inactivation of GTPases	Inactivation of GTPases	Phospholipase C

**Table 3 animals-14-02919-t003:** Methods for diagnosing clostridial myonecrosis.

Diagnostic Method	Advantage	Disadvantage
Bacterial isolation	Relatively simple execution, with media and reagents common to anaerobic bacteriology laboratories.	Time-consuming (about 48 h) and may result in inaccurate results due to the different growth ability and oxygen tolerance of histotoxic clostridia.
Direct immunofluorescence	A practical and quick protocol (about 4 h) that can be carried out directly from the imprint of the collected material.	Requires antibodies conjugated to fluorochromes and a special microscope for reading.
Immunohistochemistry	Shipping the material in formaldehyde prevents its autolysis, preventing saprophytic clostridia from multiplying. The execution is relatively simple, and the reading is carried out using light microscopy.	Relatively time-consuming, depending on histological processing prior to execution.
Multiplex PCR	Relatively simple and quick execution.	It requires the prior isolation step, adding to the disadvantages of the method and the costs of equipment and reagents.

**Table 4 animals-14-02919-t004:** Characterization of alpha, beta, epsilon and iota toxins from *C. perfringens*.

	Alpha Toxin	Beta Toxin	Epsilon Toxin	Iota Toxin
Codification	Gene *plc*	Gene *cpb*	Gene *etx*	Genes *iap* e *iab*
Origem	Chromosomal	Plasmidial	Plasmidial	Plasmidial
Molecular weight (kDa)	43	40	33.7	47.5 and 105
Main effect	Intravascular hemolysis, capillary damage and platelet aggregation.	Formation of pores and changes in vascular permeability.	Change in vascular permeability.	Change in the organization of the cellular cytoskeleton.

## Data Availability

Not applicable.

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
