# Peer review of "Clostridial Infections in Cattle: A Comprehensive Review with Emphasis on Current Data Gaps in Brazil"

_animals, 2024, doi:10.3390/ani14202919_

Round 1

Reviewer 1 Report

Comments and Suggestions for Authors

Dear Autors,

The article is written interestingly and has great potential. However, I believe it should include collected epidemiological data instead of reducing the "data gaps" to very general statements and information.

Author Response

Dear Reviewer 1,

First, I would like to thank you for your willingness to review the manuscript. Your critiques, comments, and suggestions have been carefully considered and have certainly contributed, along with the feedback from the other two reviewers, to the technical and scientific improvement of this review.

Directly addressing your question: "The article is written interestingly and has great potential. However, I believe it should include collected epidemiological data instead of reducing the 'data gaps' to very general statements and information," I would like to clarify that the main objective of this review is precisely to highlight the lack of epidemiological data on clostridial diseases in Brazil. How is it that the world’s largest beef exporter and the country with the second-largest cattle herd has so little epidemiological data? The explanation lies in the lack of specialized laboratories for diagnosing clostridial diseases. In Brazil, there are only three such laboratories: the Biological Institute of São Paulo (IB-SP), a laboratory specialized in diagnosing infectious diseases, and two laboratories from public universities (the Federal University of Minas Gerais and the Federal University of Pará, the latter of which I coordinate). These represent a statistically insufficient number to meet the demand. Another factor is the lack of training and interest among veterinarians in sending samples for diagnosis. In addition to the vast distances, the costs of sending samples by air make diagnosis unfeasible. Furthermore, most clostridial diseases have an acute clinical course, and animals are often found dead. Since necropsy must be performed within six hours after death to avoid a false-positive result—given that clostridia are commensals of the gastrointestinal tract, exist as spores in the environment, and because it is the toxins, not the bacteria, that cause the disease—all these factors complicate laboratory diagnosis and, consequently, the compilation and publication of epidemiological data.

For these reasons, we wanted to strongly emphasize the gaps in knowledge about clostridial diseases, while also sharing our experience of more than 20 years working with anaerobes, to alert the scientific community, producers, and field veterinarians about the importance of clostridial diseases and the urgent need to carry out diagnoses through sample collection and submission to laboratories. This would allow us to produce manuscripts on the occurrence, incidence, and prevalence of botulism, tetanus, enterotoxemia, and myonecrosis, not only in cattle but also in other livestock, as well as wild animals and pets.

I hope this response addresses your concerns, and I believe that from this review onwards, we will see greater epidemiological surveillance of clostridial diseases in Brazil, which will eventually result in concrete epidemiological data on their occurrence. We made modifications to the text more directly in line with the suggestions from reviewers 2 and 3, and I hope that, along with the justification I provided to you above, this will be sufficient to meet your expectations that this manuscript does indeed have the potential to be published in Animals.

Best regards,  
Felipe Masiero Salvarani

Reviewer 2 Report

Comments and Suggestions for Authors

Some additional observations and comments to be considered.

a. Great update on Clostridial diseases that is long overdue.

b. The use of Diagnostic Support services have diminished as the livestock industry matures and daily veterinary service has decreased.

c. Common risk for Clostridiosis; surgical castration, castration with Bands associated with Tetanus

d. Hepatitis caused by C hemolyticum often provoked by concomittant infection with liver flukes

e. C diffocil is sometimes diagnosed as the cause of enteric disease in foals

f. C perfringens, Type A often associated with disease in both dairy and beef replacement cattle.

g. The neurotoxins of C perfringens Type D may be responsible for the the CNS signs observed in the North American bovine disease commonly called "Nervous Coccidiosis"

h. Botulism is a ow risk disease for Canadian cattle but has been diagnosed in migrating waterfowl

i. Acute death is usually the most common way Clostridiosis presents itself. And, similarly to what the authors suggest, diagnostic support for livestock is becoming less common  

Author Response

Dear Reviewer 2,

First of all, I would like to thank you for your willingness to review the manuscript. Your criticisms, comments, and suggestions have been carefully considered, and they certainly contributed, along with those from the other two reviewers, to the technical and scientific improvement of this review. In particular, we would like to express our appreciation for your perception that: a. "Great update on Clostridial diseases that are long overdue";b. "The use of Diagnostic Support services has decreased as the livestock industry matures and daily veterinary service has decreased";i. "Acute death is usually the most common way Clostridiosis presents itself. And, similarly to what the authors suggest, diagnostic support for livestock is becoming less common."

I would also like to inform you that the main objective of this review is to demonstrate the lack of epidemiological data on Clostridial diseases in Brazil. How is it that the world’s largest beef exporter, with the second-largest cattle herd, has so little epidemiological data? The lies explanation in the lack of specialized laboratories for diagnosing Clostridial diseases. In Brazil, there are only three laboratories: the Biological Institute of São Paulo (IB-SP), a laboratory specialized in the diagnosis of infectious diseases, and two others from public universities (the Federal University of Minas Gerais and the Federal University of Pará , the latter of which is coordinated by me), which represents a number far below what is needed. Another factor is the lack of training and interest from veterinarians in sending samples for diagnosis, as the long distances and the cost of sending samples by air make the diagnosis unfeasible. Adding to all of this is the fact that most Clostridial diseases have an acute clinical course, and animals are often already found dead. Necropsies must be performed within six hours of death to avoid a false-positive result, as Clostridia are commensals of the gastrointestinal tract, present in spore form in the environment. Furthermore, it is the toxins, rather than the bacteria themselves, that are responsible for the pathological condition of the disease. All of this complicates laboratory diagnosis and, consequently, the compilation and publication of epidemiological data.
For these reasons, we wanted to highlight the gaps in the knowledge on Clostridial diseases while at the same time expressing our 20 years of experience in the field of anaerobes. This is meant to alert the scientific community, producers, and field veterinarians to the importance of Clostridial diseases and the urgent need to collect samples and send them to laboratories. Only by doing this can we produce manuscripts on the occurrence, incidence, and prevalence of botulism, tetanus, enterotoxemia, and myonecrosis, not only in cattle but also in other animals of economic interest, as well as in wildlife and pets.

Comment C: was inserted in the text (line 182).

Comment D: Indeed, hepatitis caused by C. hemolyticum is restricted to the southern region of Brazil due to the presence of liver flukes, which create pathways for C. hemolyticum. This condition is mainly associated with small ruminants, as documented in the literature. However, the same literature does not report it in cattle, which is why it was not included in the manuscript. However, I believe that with more sample submissions to diagnostic laboratories, we may obtain epidemiological data on Clostridial hepatitis in cattle in Brazil.

Comment E: C. difficile has indeed been diagnosed as responsible for enteric conditions in foals. However, it was not mentioned in the text because the review focuses on cattle, and the special topic to which the review was submitted is specific to cattle. I would like to inform you that we are writing a specialized review article on Clostridial diseases in horses, and the issue you raised will certainly be addressed.

Comment F: was inserted in the text (lines 320 and 321).

Comment G: In Brazil, coccidiosis is directly related to one of the main etiological agents of neonatal diarrhea in calves, unlike what is observed in the USA.

Comment H: In Brazil, the risk does not come from migratory wild birds as it does in Canada. Here, the issue was related to the use of so-called "chicken litter," which consists of remnants from commercial poultry production, containing feed, feces, and carcass remains. However, since 2008, the Brazilian Ministry of Agriculture, due to bovine spongiform encephalopathies, has banned the use of animal protein to feed cattle, thereby prohibiting the use of "chicken litter" in animal feed, which is why it was not discussed in the article.

I hope I have managed to address your concerns, and I believe that, starting from this review, we will have greater epidemiological surveillance of Clostridial diseases in Brazil. This could lead to concrete epidemiological data on their occurrence, confirming or refuting what we discuss in the article and what you have observed.

Best regards,

Felipe Masiero Salvarani

Reviewer 3 Report

Comments and Suggestions for Authors

The authors have not identified data and research gaps, nor the problems responsible for the clostridial infections being so common in Brazilian livestock despite the wide availability of clostridial vaccines.

Lines 135-143: I think poultry is not the main area of this manuscript as your title indicates?

Lines 242-245: Please explain the statements especially about the use of anti-tetanus serum. How is it produced and how is it available in Brazil? Also why is the tetanus vaccine so expensive in Brazil? Provide a valid reference for the statement about Brazil's FMD-free status?

Lines 354-356: Do you think 10% neutral formalin maintains the bacterial load in the sample? Otherwise, explicitly mention the use of formalin in samples for diagnostic purposes.

Lines 374-376: What problem? Mention it specifically?

Line 377: Where are these antigens produced?

Line 378: Lack of routine for what?

Lines 384-385: Please name some of these conditions?

Line 400: Replace “superacute” with “peracute”.

Lines 476-484: The authors describe how vaccination could be a possible preventive measure, but have not identified the problem why clostridial diseases could be so widespread throughout the country.

Lines 505-506: The authors mention botulism as a key clostridial disease in Brazil, but in line 116 they mention it as a sporadic disease.

Lines 519-521: Yes, but why it is not possible to administer antibiotics to the affected animals in time, especially when the disease is already prevalent in the country? And why are the costs and logistics of clostridial treatments prohibitive for their use? The most commonly used drug for clostridial infections is penicillin, which I think should not be expensive and can be transported at room temperature, so why is it unaffordable? Please explain?

Line 522: How you can recommend a more comprehensive vaccination program when in section 6 of your manuscript you already describe a wide availability and use of multivalent vaccines in the country? Where is the problem?

Lines 527-530: How could a continuous surveillance program be implemented in Brazil, and if it does not exist yet, why not?

Lines 531-542: Yes, but what exactly do you want to say?

Comments on the Quality of English Language

Mild to moderate English corrections are required.

Author Response

Dear Reviewer 3,

First, I would like to thank you for your willingness to review the manuscript. Your critiques, comments, and suggestions have been carefully considered and have certainly contributed, along with the feedback from the other two reviewers, to the technical and scientific improvement of this review.

Directly addressing your question: "The authors have not identified data and research gaps, nor the problems responsible for clostridial infections being so common in Brazilian livestock despite the wide availability of clostridial vaccines." I would like to clarify that the main objective of this review is precisely to highlight the lack of epidemiological data on clostridial diseases in Brazil. How is it that the world’s largest beef exporter and the country with the second-largest cattle herd has so little epidemiological data? The explanation lies in the lack of specialized laboratories for diagnosing clostridial diseases. In Brazil, there are only three such laboratories: the Biological Institute of São Paulo (IB-SP), a laboratory specialized in diagnosing infectious diseases, and two laboratories from public universities (the Federal University of Minas Gerais and the Federal University of Pará, the latter of which I coordinate). These represent a statistically insufficient number to meet the demand. Another factor is the lack of training and interest among veterinarians in sending samples for diagnosis. In addition to the vast distances, the costs of sending samples by air make diagnosis unfeasible. Furthermore, most clostridial diseases have an acute clinical course, and animals are often found dead. Since necropsy must be performed within six hours after death to avoid a false-positive result—given that clostridia are commensals of the gastrointestinal tract, exist as spores in the environment, and because it is the toxins, not the bacteria, that cause the disease—all these factors complicate laboratory diagnosis and, consequently, the compilation and publication of epidemiological data. I would also like to explain that despite the wide availability of vaccines, they are not purchased by farmers due to costs. In addition, there is a significant obstacle with the Brazilian government, which does not assess the quality and potency of all the immunogens produced.

For these reasons, we wanted to strongly emphasize the gaps in knowledge about clostridial diseases, while also sharing our experience of more than 20 years working with anaerobes, to alert the scientific community, producers, and field veterinarians about the importance of clostridial diseases and the urgent need to carry out diagnoses through sample collection and submission to laboratories. This would allow us to produce manuscripts on the occurrence, incidence, and prevalence of botulism, tetanus, enterotoxemia, and myonecrosis, not only in cattle but also in other livestock, as well as wild animals and pets.

I hope this response addresses your concerns, and I believe that from this review onwards, we will see greater epidemiological surveillance of clostridial diseases in Brazil, which will eventually result in concrete epidemiological data on their occurrence. We made modifications to the text more directly in line with the suggestions from reviewers 2 and 3 (you), and I hope that, along with the justification I provided to you above, this will be sufficient to meet your expectations that this manuscript does indeed have the potential to be published in Animals.

Below, I respond to your questions/suggestions/criticisms point by point:

Lines 135-143: I think poultry is not the main area of this manuscript as your title indicates?
You are correct, and this was an error on my part. I have removed the paragraph. I apologize, as I intended to discuss something related to the use of so-called "chicken litter," which consists of remnants from commercial poultry production, containing feed, feces, and carcass remains. However, since 2008, the Brazilian Ministry of Agriculture, due to concerns over bovine spongiform encephalopathy, has banned the use of animal protein in cattle feed, thereby prohibiting the use of "chicken litter" in animal feed, which is why it was not discussed in the article. I ended up forgetting to remove it entirely. Thank you for pointing that out.

Lines 242-245: "Please explain the statements especially about the use of anti-tetanus serum. How is it produced and how is it available in Brazil?" Serums are used to treat intoxications caused by the venom of venomous animals or by toxins from infectious agents, such as those responsible for diphtheria, botulism, and tetanus. The production of serum follows these steps:

The lyophilized antigen is diluted and injected into a horse in appropriate doses. This process takes 40 days and is called hyperimmunization. After hyperimmunization, an exploratory bloodletting is performed, taking a blood sample to measure the level of antibodies produced in response to the antigen injections. When the antibody level reaches the desired level, a final bloodletting is performed, extracting about fifteen liters of blood from a 500 kg horse in three stages, with 48-hour intervals. Antibodies are found in the plasma (the liquid part of the blood). The serum is obtained by purifying and concentrating this plasma. The red blood cells (which make up the red part of the blood) are returned to the animal using a technique developed at the Butantan Institute called plasmapheresis. This replenishment technique reduces the side effects caused by the bloodletting.
At the end of the process, the serum obtained is subjected to quality control tests: 6.1. biological activity – to verify the amount of antibodies produced; 6.2. sterility – to detect any contamination during production; 6.3. safety – a test to ensure human use safety; 6.4. pyrogen test – to detect the presence of substances that can cause temperature changes in patients; and 6.5. physical-chemical tests. Hyperimmunization for serum production has been performed in horses since the beginning of the century because they are large animals and produce a high volume of antibody-rich plasma for the industrial processing of serum to meet national demand, without harming the animals in the process. These horses are under medical-veterinary supervision and receive a richly balanced diet. The plasma processing for serum production is carried out in a closed system entirely developed by the Butantan Institute, which aims to produce 600,000 serum vials annually, meeting the quality control and biosafety standards of the World Health Organization. The serum is then marketed to producers and veterinarians. (Here is the webpage with information about Brazilian tetanus antitoxin serum: https://butantan.gov.br/assets/arquivos/soros-e-vacinas/soros/Soro%20antitet%C3%A2nico.pdf)

"Also, why is the tetanus vaccine so expensive in Brazil?" This is a market issue, as there are few pharmaceutical companies that sell the tetanus vaccine for animals, and the price is based on free-market competition.

"Provide a valid reference for the statement about Brazil's FMD-free status?" Portaria nº 678 de 30 de Abril de 2024 - BRASIL - Zona Livre de Febre Aftosa Sem Vacinação. The Ministry of Agriculture, Livestock, and Supply (MAPA) recognizes Brazil as an FMD-free zone without vaccination. In summary, with the end of the last immunization campaign against foot-and-mouth disease for 12 Brazilian states and part of the state of Amazonas, Brazil advances in the Strategic Plan of the National Foot-and-Mouth Disease Eradication Program (PNEFA) and becomes entirely free of the disease without vaccination. This action, which is part of the process for international recognition by the World Organization for Animal Health (WOAH), marks the end of a vaccination cycle that began over 50 years ago and highlights the quality of national livestock production and the strength of the Official Veterinary Service. By eradicating foot-and-mouth disease and consolidating itself as a country free of the disease without vaccination, Brazil strengthens its position in the international market, increasing consumer and trade partner confidence in the quality and safety of Brazilian animal products. The international recognition of the country’s FMD-free status without vaccination is granted by WOAH. For this, the Organization requires the suspension of foot-and-mouth disease vaccination and the prohibition of entry of vaccinated animals into the states for at least 12 months. Brazil plans to submit its request for recognition to the World Organization for Animal Health in August 2024. If approved, the result will be presented in May 2025 during the organization’s General Assembly.

Therefore, there is no official reference yet, but Brazil no longer vaccinates its cattle and buffalo against FMD, and since August 2024, the vaccine is no longer marketed or produced in the country. (The two sources where you can access this information are: https://www.gov.br/agricultura/pt-br/assuntos/noticias/brasil-se-torna-livre-de-febre-aftosa-sem-vacinacao and https://www.in.gov.br/en/web/dou/-/portaria-mapa-n-678-de-30-de-abril-de-2024-557403229)

Lines 354-356: Do you think 10% neutral formalin maintains the bacterial load in the sample? Otherwise, explicitly mention the use of formalin in samples for diagnostic purposes.

No, 10% neutral formalin does not maintain the bacterial load in a sample. Formalin is a fixative, commonly used to preserve tissue morphology for histological examination, but it is not suitable for preserving bacterial viability. In fact, formalin will inactivate bacteria, making it impossible to culture or perform molecular diagnostics like PCR from the sample. For diagnostic purposes, especially when bacterial identification or culturing is required, samples should be collected and preserved in sterile conditions without formalin. Was inserted in manuscript (lines 357 e 356) that formalin is used for preserving tissue morphology and is not suitable for maintaining bacterial load in samples intended for microbial diagnostics. Thank you for pointing that out.

Lines 374-376: What problem? Mention it specifically? Lack of diagnosis and epidemiological data on clostridial diseases in Brazil. Inserted in the text (line 379).

Line 377: Where are these antigens produced? The antigens used in vaccine production in Brazil are typically produced in specialized laboratories, including public and private research institutions. In the case of vaccines against clostridial diseases, for example, the antigens are often produced by biological institutes like the Biological Institute of São Paulo (IB-SP) and public university laboratories, such as those at the Federal University of Minas Gerais and Federal University of Pará. Additionally, some antigens are produced by private pharmaceutical companies that specialize in veterinary vaccines.

Line 378: Lack of routine for what? However, the lack of routine use of clostridial vaccines impedes the prevention of these diseases. Inserted in manuscript.

Lines 384-385: Please name some of these conditions? Enterotoxemias caused by C. perfringens occur under specific conditions in the presence of certain predisposing factors, such as abrupt changes in diet, overeating, intestinal stasis, or any condition that disrupts the normal gut flora, allowing the bacteria to proliferate and produce toxins that lead to the disease. Inserted in manuscript (lines 388 -390).

Line 400: Replace “superacute” with “peracute”. Change made.

Lines 476-484: The authors describe how vaccination could be a possible preventive measure, but have not identified the problem why clostridial diseases could be so widespread throughout the country. Clostridial diseases are widespread throughout Brazil and other regions all the world, primarily because Clostridium species are spore-forming bacteria, which allows them to persist in the environment for extended periods. The spores are highly resistant to environmental factors such as heat, desiccation, and disinfectants, facilitating their distribution in soil, water, and animal feces across the country. In Brazil, where livestock farming is extensive, environmental contamination with Clostridium spores is common. Furthermore, the country's vast geographical size, combined with factors such as variable climate conditions, poor vaccination coverage, and limited access to diagnostic resources, exacerbates the spread of these diseases. The lack of widespread vaccination and difficulties in ensuring effective administration of vaccines in remote areas contribute to the prevalence of clostridial diseases in livestock. Additionally, Brazil's high livestock population increases the likelihood of environmental contamination, making clostridial infections more frequent.

Would the reviewer prefer that I insert this paragraph I responded to in the text rather than the way it appears in the manuscript?

Thus, while vaccination is a crucial preventive measure, addressing the problem of the bacteria’s persistence in the environment and improving access to vaccination and diagnostics are essential to reduce the incidence of these diseases.

Lines 505-506: The authors mention botulism as a key clostridial disease in Brazil, but in line 116 they mention it as a sporadic disease. I believe there is a misinterpretation, as I did not mean sporadic because it is not important, but rather the form of presentation of the disease, since we have endemic and sporadic botulism.

I will try to explain in a more didactic way:

The difference between endemic and sporadic botulism in cattle lies in the frequency and geographic distribution of the disease:

  1. Endemic Botulism: This refers to regions where botulism occurs regularly and consistently over time, often due to environmental factors that favor the persistence of Clostridium botulinum spores. In certain areas of Brazil, particularly in regions with poor pasture conditions, phosphorus-deficient soils, and inadequate feed management, botulism is endemic. Cattle in these areas may be prone to ingesting contaminated material, such as decomposing carcasses or spoiled feed, leading to recurrent outbreaks. Thus, in such regions, botulism is considered a persistent problem.
  2. Sporadic Botulism: Sporadic cases, on the other hand, occur infrequently and are less predictable. In some parts of Brazil, botulism is not consistently observed and only appears occasionally, possibly due to isolated exposure events, such as accidental ingestion of toxins from contaminated feed or water sources. In these areas, botulism is not a regularly recurring issue, and outbreaks are often unexpected.

In summary, while botulism is a major concern in Brazil, it can be endemic in certain regions due to environmental and management factors, but it is considered sporadic in other areas where outbreaks occur unpredictably.

Lines 519-521: Yes, but why it is not possible to administer antibiotics to the affected animals in time, especially when the disease is already prevalent in the country? And why are the costs and logistics of clostridial treatments prohibitive for their use? The most commonly used drug for clostridial infections is penicillin, which I think should not be expensive and can be transported at room temperature, so why is it unaffordable? Please explain?

Clostridial diseases, such as botulism, tetanus, and enterotoxemia, are caused primarily by toxins produced by the bacteria, not the bacteria themselves. This means that by the time clinical signs appear, the bacterial toxins have already caused significant damage, making treatment with antibiotics less effective. Clostridium species, being spore-forming anaerobes, can persist in the environment for extended periods without causing disease until specific conditions trigger toxin production. Therefore, the challenge is not just eliminating the bacteria but neutralizing the toxins they produce, which requires rapid intervention.

  1. Timing and Efficacy of Antibiotics: The administration of antibiotics, such as penicillin, is only effective if given early in the disease course, ideally before toxins have caused extensive damage. In many cases of clostridial diseases, affected animals are often found already severely ill or dead due to the rapid progression of the disease. This makes timely administration of antibiotics difficult, especially in extensive farming systems where animals may not be observed regularly, or where access to veterinary services is delayed.
  2. Prohibition of Indiscriminate Use of Antibiotics: Indiscriminate or preventive use of antibiotics is not a viable strategy for controlling clostridial diseases, as it could lead to antimicrobial resistance, which is a growing global concern. Moreover, routine antibiotic use would result in antibiotic residues in meat and milk, which is heavily regulated due to one health concerns. These issues make preventive antibiotic use impractical and discouraged.
  3. Cost and Logistics of Treatment: While penicillin itself may not be prohibitively expensive, the cost of treating a large number of animals on a commercial farm can become significant when combined with the logistics of regular monitoring, veterinary services, and administering antibiotics. Additionally, clostridial diseases often occur in extensive, remote regions of Brazil, where access to veterinary care and medical supplies is limited. Transporting antibiotics and other medical supplies to these areas, combined with the labor required to treat animals, adds to the overall cost. The fact that the disease can progress quickly and affect multiple animals in a herd at once can make treatment unfeasible, both logistically and financially, for many farmers.

In conclusion, the combination of rapid disease progression, the need for early intervention, the risks of antibiotic overuse, and the logistical challenges in rural Brazil makes reliance on antibiotics an impractical solution for clostridial diseases. This is why vaccination and other preventive measures are strongly emphasized as the most effective approach.

Line 522: How you can recommend a more comprehensive vaccination program when in section 6 of your manuscript you already describe a wide availability and use of multivalent vaccines in the country? Where is the problem?

While section 6 of the manuscript describes the wide availability of multivalent vaccines against clostridial diseases in Brazil, several challenges limit the effectiveness and coverage of vaccination programs in the country:

  1. Vaccine Availability vs. Accessibility: Although vaccines are widely available, their actual usage is inconsistent, particularly in remote or less-developed regions. Many small and medium-sized farmers may not have easy access to vaccines due to logistical challenges such as the distance from veterinary suppliers, inadequate storage facilities (e.g., for maintaining the cold chain), and limited access to veterinary services. These logistical barriers result in lower vaccination coverage than expected, particularly in extensive cattle farming systems common in Brazil.
  2. Economic Constraints: The cost of vaccines, though generally affordable, is still a barrier for many farmers, especially those with large herds or low profit margins. Additionally, the indirect costs associated with vaccination programs, such as labor, transportation, and veterinary services, can be prohibitive. As a result, some producers opt not to vaccinate their livestock, which leaves gaps in herd immunity and contributes to sporadic outbreaks of clostridial diseases.
  3. Perception and Knowledge Gaps: There is also a challenge related to the awareness and understanding of the importance of clostridial vaccination among farmers. In some cases, farmers may not recognize the need for regular vaccinations, particularly for diseases that they have not experienced recently or are not well-educated on. Furthermore, because many clostridial diseases have acute clinical courses and sudden deaths, producers may not associate the disease with a lack of vaccination, especially if they do not seek a laboratory diagnosis.
  4. Vaccine Efficacy and Potency Issues: As mentioned in section 6, another critical problem is that the Brazilian government does not evaluate the quality and potency of all vaccines produced and marketed in the country. This can result in variations in vaccine efficacy, with some farmers experiencing inadequate protection even when they vaccinate their herds. Improving the regulation and quality control of these vaccines is essential to ensure effective immunization.

In summary, while multivalent vaccines are available in Brazil, there are significant gaps in access, farmer adherence, and vaccine quality control. Therefore, a more comprehensive vaccination program should focus not only on the availability of vaccines but also on improving accessibility, farmer education, and the quality and efficacy of the vaccines administered.

Lines 527-530: How could a continuous surveillance program be implemented in Brazil, and if it does not exist yet, why not?

Implementing a continuous surveillance program for clostridial diseases in Brazilian cattle would require significant investment in infrastructure, education, and coordination between government agencies, veterinarians, and livestock producers. While such programs exist in certain countries with smaller geographic areas and more centralized livestock industries, Brazil faces unique challenges that have hindered the establishment of a nationwide clostridial disease surveillance program.

  1. Geographical Size and Agricultural Practices: Brazil is a vast country with large, extensive cattle farming regions, particularly in the Amazon, Pantanal, and Cerrado biomes. Cattle are often raised on large, remote ranches with minimal direct human supervision, making routine monitoring of animal health and early detection of disease difficult. This extensive farming model complicates sample collection and reporting, especially for diseases with rapid progression like clostridial infections.

  1. Lack of Diagnostic Infrastructure: Brazil currently has a limited number of specialized laboratories capable of diagnosing clostridial diseases. As mentioned earlier, only a few laboratories, including those at the Biological Institute of São Paulo (IB-SP), the Federal University of Minas Gerais, and the Federal University of Pará, are equipped to perform such diagnostics. This number is insufficient to meet the demand, particularly in remote areas where cattle ranching is most common. The logistical challenges of transporting samples across vast distances and maintaining the quality of these samples add to the difficulties.

  1. Veterinary Resources and Farmer Awareness: Many veterinarians in rural areas are not adequately trained or incentivized to collect and send samples for clostridial disease diagnostics. Additionally, many farmers lack awareness about the importance of disease surveillance and are reluctant to spend money on diagnostic services, particularly for diseases like clostridiosis, which can present as sudden deaths with minimal warning. The cost of sending samples for diagnosis and the relatively low engagement of field veterinarians contribute to the absence of a formal surveillance system.

  1. Government Priorities and Funding: While the Brazilian Ministry of Agriculture, Livestock, and Supply (MAPA) has made significant efforts in disease control programs, such as for foot-and-mouth disease (FMD), clostridial diseases have not received the same level of attention. This is partly because they are often considered endemic and primarily managed through vaccination rather than requiring continuous surveillance. Establishing a surveillance program would require coordinated government support, significant financial investment, and the creation of a nationwide reporting system, which is not currently in place.

Why such a program doesn't exist yet:

The combination of Brazil's geographic size, the extensive nature of cattle farming, limited diagnostic infrastructure, and the absence of consistent government prioritization for clostridial diseases has made it challenging to implement a continuous surveillance program. Additionally, the financial and logistical barriers to regularly monitoring large cattle populations in remote areas further complicate the feasibility of such a system.

How a program could be implemented:

A surveillance program could be developed through:

- Increased investment in diagnostic infrastructure: Expanding the number of laboratories and improving access to diagnostic services in rural regions.

- Veterinarian and farmer education: Training programs to raise awareness about the importance of reporting cases and collecting samples for diagnosis.

- Government involvement and incentives: MAPA could create incentive-based programs to encourage farmers and veterinarians to participate in disease surveillance and reporting.

- Technological solutions: Implementing mobile diagnostic tools and digital platforms for real-time reporting and monitoring could make it more feasible to track clostridial disease outbreaks across the country.

In summary, while challenges exist, a coordinated effort involving the government, farmers, veterinarians, and enhanced infrastructure could make a continuous clostridial disease surveillance program in Brazil feasible.

Lines 531-542: Yes, but what exactly do you want to say? Rewrite the paragraph in an attempt to make it clearer for the reader and for this reviewer, see if it meets your expectations.

As future perspectives are expected prioritize the development of more effective and long-lasting vaccines. Specifically, research into multivalent vaccines, which could protect against multiple clostridial species with a single injection, could greatly improve vaccine compliance among farmers and lead to better disease control outcomes. In addition to vaccines, advancements in diagnostic technologies are essential. The development of rapid, on-farm diagnostic tests would enable earlier detection of clostridial diseases, allowing for timely intervention and improving animal survival rates. Portable and cost-effective diagnostic tools would be particularly beneficial for small-scale and remote farmers, who often face logistical challenges in accessing veterinary services. Education and training are also critical. Farmers need to be equipped with the knowledge of best practices for disease prevention and management. Extension services and veterinary outreach programs can play a pivotal role in spreading this knowledge. Moreover, ongoing research into the epidemiology, pathogenesis, and control of clostridial diseases is essential to gaining new insights and developing innovative solutions. Strong collaboration between research institutions, government agencies, and the private sector will be key to driving advancements in clostridial disease management in Brazil. This revision clarifies that your goal is to improve disease prevention through enhanced vaccines, diagnostics, farmer education, and collaboration across sectors.

Best regards, 

Felipe Masiero Salvarani

Round 2

Reviewer 1 Report

Comments and Suggestions for Authors

Dear Autors,

the article is a review paper. The sparse epidemiological data, or rather the almost complete lack of it concerning diseases caused by Clostridium sp. in Brazil, do not contribute anything to the article. I therefore suggest changing the title from 'Clostridial infections in cattle: a comprehensive review with emphasis on current data gaps in Brazil' to simply 'Clostridial infections in cattle' and removing the sections on the epidemiology of these diseases in Brazil from the article.

Author Response

Dear Reviewer 1,

Thank you for reviewing the manuscript again. While I appreciate your feedback, I must respectfully disagree with your suggestions. You stated: "suggest changing the title from 'Clostridial infections in cattle: a comprehensive review with emphasis on current data gaps in Brazil' to simply 'Clostridial infections in cattle' and removing the sections on the epidemiology of these diseases in Brazil from the article." However, this is not a minor revision, but rather a fundamental change to the structure and purpose of the article, which I will explain below.

You emphasize that the article contains "sparse epidemiological data, or rather the almost complete lack of it concerning diseases caused by Clostridium sp. in Brazil" and suggest "removing the sections on the epidemiology of these diseases in Brazil from the article." However, of the 54 total references, 28 (more than 50%) specifically address clostridial diseases in Brazil, with nearly all of them focusing on the epidemiology of these diseases. Thus, removing this information would eliminate the central focus and objective of the article, which is precisely to highlight these data gaps in Brazil.

I would like to clarify that the main objective of this review is precisely to highlight the lack of epidemiological data on clostridial diseases in Brazil. How is it that the world’s largest beef exporter and the country with the second-largest cattle herd has so little epidemiological data? The explanation lies in the lack of specialized laboratories for diagnosing clostridial diseases. In Brazil, there are only three such laboratories: the Biological Institute of São Paulo (IB-SP), a laboratory specialized in diagnosing infectious diseases, and two laboratories from public universities (the Federal University of Minas Gerais and the Federal University of Pará, the latter of which I coordinate). These represent a statistically insufficient number to meet the demand. Another factor is the lack of training and interest among veterinarians in sending samples for diagnosis. In addition to the vast distances, the costs of sending samples by air make diagnosis unfeasible. Furthermore, most clostridial diseases have an acute clinical course, and animals are often found dead. Since necropsy must be performed within six hours after death to avoid a false-positive result—given that clostridia are commensals of the gastrointestinal tract, exist as spores in the environment, and because it is the toxins, not the bacteria, that cause the disease—all these factors complicate laboratory diagnosis and, consequently, the compilation and publication of epidemiological data.

For these reasons, we wanted to strongly emphasize the gaps in knowledge about clostridial diseases, while also sharing our experience of more than 20 years working with anaerobes, to alert the scientific community, producers, and field veterinarians about the importance of clostridial diseases and the urgent need to carry out diagnoses through sample collection and submission to laboratories. This would allow us to produce manuscripts on the occurrence, incidence, and prevalence of botulism, tetanus, enterotoxemia, and myonecrosis, not only in cattle but also in other livestock, as well as wild animals and pets.

For these reasons, I must disagree with your suggestion to modify the title and remove sections on Brazilian epidemiological data.

Best regards,  

Felipe Masiero Salvarani

Reviewer 3 Report

Comments and Suggestions for Authors

I think the manuscript is sufficiently revised and can be recommended for publication.

Comments on the Quality of English Language

Minor English revisions...

Author Response

Dear Reviewer 3,
Thank you for once again reviewing the manuscript and reaching the conclusion that: "I think the manuscript is sufficiently revised and can be recommended for publication."
Best regards
Felipe Masiero Salvarani
